# Long-Term Monitored Norway Spruce Plots in the Ore Mountains—30 Years of Changes in Forest Health, Soil Chemistry and Tree Nutrition after Air Pollution Calamity

**DOI:** 10.3390/plants13172379

**Published:** 2024-08-26

**Authors:** Radek Novotný, Věra Fadrhonsová, Vít Šrámek

**Affiliations:** Forestry and Game Management Research Institute (FGMRI), Department of Forest Ecology, Strnady 136, 25202 Jíloviště, Czech Republic; fadrhonsova@vulhm.cz (V.F.); sramek@vulhm.cz (V.Š.)

**Keywords:** air pollution, forest soils, nutrition, defoliation, Norway spruce

## Abstract

The Ore Mountains were historically one of the most polluted areas in Europe, where high sulphur dioxide concentrations and a high level of atmospheric deposition led to a vast decline in Norway spruce stands in the mountain ridge plateau. In this article, we evaluate the trends in the atmospheric deposition load, soil chemistry, tree nutrition, crown defoliation and height increment in a network of twenty research plots monitored for last thirty years in this region. The decrease in sulphur and nitrogen deposition was most pronounced at the end of 1990s. Extreme values of sulphur deposition (100–200 kg.ha^−1^.year^−1^) were recorded in throughfall under mature Norway spruce stands in the late 1970s, and after felling of the damaged stand, the deposition levels were comparable to open plot bulk deposition. Nitrogen deposition decreased more slowly compared with sulphur, and a decrease in base cation deposition was observed concurrently. The current deposition load is low and fully comparable to other mountain areas in central Europe. Accordingly, the health of young spruce stands, as assessed by defoliation and height increment, has improved and now corresponds to the Czech national average. On the other hand, no significant changes were observed in the soil chemistry, even though some of the plots were limed. Acidic or strongly acidic soil prevails, often with a deficiency of exchangeable calcium and magnesium in the mineral topsoil, as well as decreases in available phosphorus. This is reflected in the foliage chemistry, where we see an imbalance between a relatively high content of nitrogen and decreasing contents of phosphorus, potassium and calcium. Despite the observed positive trends in air quality and forest health, the nutritional imbalance on acidified soils poses a risk for the future of forest stands in the region.

## 1. Introduction

Forest damage has occurred in Europe since the Middle Ages, especially in connection with the development of mining and ore processing [1,2]. The pressure of air pollution on ecosystems has increased since the beginning of industrial times, i.e., approximately since the middle of the 19th century [3,4,5]. Since the beginning of the 20th century, sulphur (S) emissions and deposition have increased steadily [6,7], and sulphate deposition in forested ecosystems of northern and central Europe peaked in the early 1980s, reaching loads of more than 100 kg S.ha^−1^.year^−1^ at the most polluted sites [8,9]. High concentrations of sulphur dioxide (and other phytotoxic elements, such as fluorine) in the air directly damage tree foliage and lead to forest decline in huge areas [1,10,11]. High atmospheric deposition loads also negatively affect soil conditions. This was, and still is, manifested by decreases in pH values in both the humus layer and mineral soil, causing important elements (Ca, Mg, K) to be leached from the rooting zone and mobilisation of potentially toxic elements, especially aluminium [12,13,14].

A heavily polluted area in Europe is the so-called black triangle, a region on the borders between the Czech Republic, Germany and Poland. The main sources of pollution are coal-combusting power plants that generate heat and power and the chemical industry, which also uses brown coal [15]. The pollution load peaked during the second half of the 1970s and first half of the 1980s, and more than 40,000 ha of forest stands died due to high concentrations of sulphur and its compounds in the ambient air during this time [2].

As a reaction to unexpected forest decline in the 1970s, which affected not only the region of the black triangle, the International Co-operative Programme on Assessment and Monitoring of Air Pollution Effects on Forests (ICP Forests) was launched under the Convention on Long-Range Transboundary Air Pollution (CLRTAP) from the United Nations Economic Commission for Europe (UN-ECE). ICP Forests had one main objective: to provide a periodic overview of the spatial and temporal variation in forest conditions in relation to anthropogenic and natural stress factors (in particular, air pollution) by means of European-wide and national large-scale representative monitoring via a systematic network [16]. The network was established predominantly in mature forest stands (>60 years old). In regions with widespread forest decline and partial deforestation, there were, however, not suitable (mature) forest stands for establishing these monitoring plots. Therefore, plots in young Norway spruce stands up to 30 years were established in these areas [17] to monitor and evaluate the condition and development of the forest stands after the air pollution disaster.

The aim of this article was to summarise and assess thirty years of monitoring data from young Norway spruce stands on the ridge of the Ore Mountains, describe how the forests appear and determine whether the forest ecosystem is currently in good condition.

## 2. Results

### 2.1. Atmospheric Deposition

In the plot called Moldava, throughfall deposition of sulphur (S-SO_4_^2−^) reached a yearly rate of 190 kg.ha^−1^.year^−1^ in the mature Norway spruce stand in 1979. After the stand was cut down at the beginning of the 1980s due to high mortality of the trees, the deposition rate decreased to ca 50 kg.ha^−1^.year^−1^. In the open plot, sulphur bulk deposition decreased from approximately 40 kg.ha^−1^.year^−1^ at the beginning of the 1980s to approximately 20 kg.ha^−1^.year^−1^ in the 1990s (after desulphurisation of power plants). In the last five years (2018–2022), the yearly values of S-SO_4_^2−^ deposition ranged from 2.9 to 4.0 kg.ha^−1^.year^−1^ for throughfall and from 2.5 to 3.3 kg.ha^−1^.year^−1^ for bulk deposition. At the end of the 1970s, the yearly deposition of nitrogen (sum of N-NO_3_^−^ and N-NH_4_^+^) reached 35 kg.ha^−1^.year^−1^ in the spruce stand and 21 kg.ha^−1^.year^−1^ in the open plot, decreasing gradually during the measurement period to 4–7 kg.ha^−1^.year^−1^ at both localities, which was slightly higher than the throughfall (Figure 1). A decreasing trend was also observed in the case of base cation deposition (Ca, Mg) due to reductions in dust particles and ash emissions. The deposition of base cations was the only source of these nutrients, except for weathering (Figure 2). Annual calcium deposition decreased significantly during the measurement period from 90.2 to 0.5 kg.ha^−1^.year^−1^ in the throughfall, and from 40.6 to 2.3 kg.ha^−1^.year^−1^ in the open plot. The magnesium deposition decreased from 8.6 to 1.7 kg.ha^−1^.year^−1^ in the throughfall, and from 4.5 to 1.1 kg.ha^−1^.year^−1^ in the open plot. Recently, the deposition of these elements has been relatively stable.

### 2.2. Soil Chemistry

For soil chemistry, there are available data from seven sampling surveys in four-year intervals for the organic layer (FH horizon) and mineral soil down to a 30 cm depth. The pH(H_2_O) values in individual sampling years ranged between 3.69 and 6.46 in the organic horizon and between 3.73 and 6.06 in the mineral soil. The pH(KCl) values ranged between 2.76 and 5.70 in the organic horizon and between 2.75 and 4.90 in the mineral soil (Figure 3). Following the national evaluation limits [18], soils could be characterised as acidic (pH(KCl) between 4 and 5) or strongly acidic (pH(KCl) lower than 4). More than 80% samples of both layers had pH values lower than 4. Evaluating the entire 25-year investigation period, pH values (both active and exchangeable) did not vary during that time; only slight changes were observed, without any increasing trend, following the rapid decrease in acid deposition.

The total nitrogen content varied between 0.70 and 2.48% in the organic horizon and between 0.08 and 0.98% in the mineral soil, which means that the N content in mineral topsoil was still high (Figure 4). Among the mineral soil samples, 71% exhibited an increased N content that was higher than 0.2%, which is considered to be an elevated value.

The content of exchangeable base cations (Ca, Mg, K) in the organic layer (FH horizon) ranged from 184.2 to 8321 mg.kg^−1^ for calcium, from 65.1 to 2238 mg.kg^−1^ for magnesium and from 85.1 to 1112 mg.kg^−1^ for potassium. In the mineral soil down to a 30 cm depth, the concentrations of exchangeable base cations ranged from 6.0 to 2458 mg.kg^−1^ for Ca, from 4.5 to 415 mg.kg^−1^ for Mg and from 14.6 to 396 mg.kg^−1^ for K (Figure 5). According to the nutrition limits classification, the base cation content in the organic layer was quite sufficient, with only five samples exhibiting a low calcium content. A worse situation was observed in the mineral soil, with 13% of samples having a low content, 68% of samples having a very low content of exchangeable Ca, 68% of samples having a low content of exchangeable Mg and 37% having a low content of exchangeable K. A higher content of exchangeable base cations was observed at the localities where liming or fertilising was performed. The effect of liming was apparent especially in the organic horizon, where we observed a temporal increase in exchangeable base cation content; nevertheless, in the mineral topsoil, changes in base cation content were very small.

The available phosphorus content was seriously low, ranging from 2.80 to 457.2 mg.kg^−1^, with a median of 26.6 mg.kg^−1^, in the organic layer, and from 0.4 to 229.6 mg.kg^−1^, with a median of 5.88 mg.kg^−1^, in the mineral soil. More than 80% of the organic horizon samples and 88% of the mineral soil samples exhibited a very low (<20 mg.kg^−1^) contents of this element; most of the mineral soil samples did not even reach 10 mg.kg^−1^ (Figure 6). However, the total P content in soil was relatively high, and its supply in an available form in the soil could be replenished.

### 2.3. Foliage Chemistry

The nitrogen content in the needles varied between 10 and 22 g.kg^−1^ in current-year needles and between 8 and 20 g.kg^−1^ in one-year-old needles. In current-year needles, the median N content was always above the limit of deficiency (13 g.kg^−1^), and in one-year-old needles, only two years had a median value below this limit. We observed a slight increase in N content, but there was fluctuation in the registered values during the period of investigation (Figure 7).

The phosphorus content varied between 0.8 and 3.1 g.kg^−1^ in current-year needles and between 0.8 and 2.6 g.kg^−1^ in one-year-old needles. There was a clear decreasing trend in P content in both needle classes. We registered values below the limit of deficiency (1.2 g.kg^−1^) much more often during the second half of the investigation period (Figure 8). A slight increase in N content and a decrease in P content led to an imbalanced ratio between these two important nutrients.

The calcium content varied between 1 and 8 g.kg^−1^ in current-year needles and between 2 and 12 g.kg^−1^ in one-year-old needles (Figure 9). Calcium is strongly bonded in cell walls, and its content usually increases from younger to older needle classes. We registered a slight decrease in Ca content throughout the investigation period in both current-year and one-year-old needles.

The potassium content varied between 3 and 11 g.kg^−1^ in current-year needles and between 3 and 10 g.kg^−1^ in one-year-old needles (Figure 10). A decrease in K content throughout the investigation period was observed, especially during the second half of this period. Potassium is an important nutrient for water regulation (stomata conductance) and plays a key role in frost resistance. Thus, its decrease may pose a risk for trees. 

The magnesium content varied between 0.3 and 2.0 g.kg^−1^ in current-year needles and between 0.3 and 2.2 g.kg^−1^ in one-year-old needles. Mg is a mobile nutrient, and trees can transfer it from older to younger needles, which are most active in photosynthesis. The Mg content was lower at the beginning of the investigation, but the situation has improved since 2000. Nevertheless, in some plots, especially older needles still exhibit an Mg content below the limit of deficiency (0.7 g.kg^−1^) in some years (Figure 11).

Sulphur and its compounds were one of the reasons for forest decline during the 20th century, and its concentration at the beginning of the investigation was often higher than 2 g.kg^−1^ (Figure 12). After desulphurisation of the main pollution sources, the situation improved quite quickly. The median S content varied between 1.1 and 1.4 g.kg^−1^ in both evaluated needle classes since 2001. 

Similar trends were observed with fluorine as the brown coal used in this region is relatively rich in fluorine content (Figure 13). In the last two decades of investigation, its content typically remained below 2 mg.kg^−1^.

### 2.4. Forest Health—Defoliation and Height Increment

The median value of defoliation exceeded 40% from 1995 to 2000, with high variability for individual trees, and needle loss ranging from 20% to more than 70% in the same plot. From 2000 to 2008, the median of defoliation was around 30%; after 2009, the median of defoliation decreased to 20% or less (Figure 14). In the last decade, dead or almost dead trees were found due to heat waves and drought episodes, but the general health state, as expressed by crown defoliation, was significantly improved in comparison with the second half of the 1990s.

The median value of height increment varied mainly between 40 and 70 cm.year^−1^ during the investigation period. Most often, it was around 60 cm.year^−1^, which is a really large yearly step, especially for Norway spruces older than 40 years that grow in mountain areas higher than 800 m a. s. l. Under 50 or 40 cm, the yearly height increments decreased only after 2013. However, in some plots, yearly height increments have exceeded 80 cm since 2013. The decrease in height increment from 1996 to 1998 was caused by heavy crown damage due to frost/rime during the winter of 1995/1996, and recovery of the usual rate of height increment took three years (Figure 15).

## 3. Discussion

The Ore Mountains are the region where the most severe influence of air pollution on forests was documented. This pollution resulted in forest dieback on more than 40,000 ha during the 1970s and 1980s [19,20]. During the 1990s, pollution decreased significantly due to de-sulphurisation of the major sources and a decrease in industrial production in the region [21]. The air concentration of sulphur dioxide dropped rapidly by tenfold between 1989, with a mean yearly concentration of ≥50 μg.m^−3^, and 1999 when there was a yearly SO_2_ concentration of around 5 μg.m^−3^ [1]. As our data describe, atmospheric deposition has been gradually decreasing, with sulphur decreasing more rapidly than total nitrogen. This is in line with local [5,15,22,23] and broader European [24,25,26,27] studies. The difference between bulk deposition and throughfall was extreme, especially for sulphur, during the peak pollution period in 1978 and 1980, with S under the mature Norway spruce stand exceeding 200 kg.ha^−1^.year^−1^. After the forest was felled due to tree mortality, the atmospheric deposition decreased, and even after reforestation by substitute tree species (mountain ash), was comparable to the open plot levels. This “enhancing” effect provided by forest stands possibly led to the increased deposition load to forest soil in the western part of the Ore Mountains, where spruce trees were not as damaged and were not felled in large areas, unlike in the eastern Ore Mountains [28].

As expressed by crown defoliation, the reaction of trees to better air quality is clearly visible during the five-year period (1996–2000) after the last direct air pollution damage during the winter season of 1995/96 [29,30]. The decrease in defoliation continued over the next ten years, and in the last decade, it was about 20% (Figure 14), which is similar to or even less than the national average defoliation for Norway spruce in the Czech Republic [31].

Although air quality improved significantly during the decade following steps to achieve air pollution reduction, the soil chemistry and forest nutrition levels have changed very slowly. The results from our plots were partly influenced by forest liming, which was realised in a part of the region [32], but still did not exhibit positive trends in acidity or nutrient content. In contrast, the decreasing content of available phosphorus is clearly visible. One study [33] evaluated soil chemistry in long-term monitoring plots in Lower Saxony and confirmed that the soil recovery process (based on increases in pH values and base saturation) was very slow and still rather reversed or not changing. Moreover, in soils under coniferous stands, the recovery is slower due to the higher amounts of sulphur stored in soil (higher deposition input). High sulphur accumulation during the period of high deposition rates can cause higher output from these polluted soils for many years after decreasing the high deposition load [34]. Similar results have been reported [35] in Austria, where a higher sulphur content in the topsoil 30 years after desulphurisation than that before desulphurisation was observed. They concluded that a major part of historically deposited sulphate was still cycling via plant uptake and litter fall through the organic S pool. Very similar results have been observed in Switzerland, where slow and weak reactions of soil and soil solution chemistry that decrease S deposition were observed [27].

From the Ore Mountains, there have been repeated reports of changes connected with malnutrition, particularly for magnesium, but also in relation to the concentrations of other nutrients in soil and/or in Norway spruce needles influenced by acid deposition [9,11,23,30,36]. With a decrease in acid deposition, the deposition of base cations, mainly calcium and magnesium, has decreased as a result of the strong reductions in dust and ash emissions. In addition to weathering and litter decomposition, deposition is a source of base cations, mainly Ca and Mg [37]. The recovery process of soil is dependent on deposition load, deposition of base cations and forest management [38]. A study from another heavy polluted area of the black triangle region (Jizerské Mountains) showed that recovery of forest soil under spruce and beech was possible but took many decades. In this case, signs of soil recovery were detectable after more than 30 years [39]. One possible way to compensate for acid input to the forest ecosystem is fertilising or liming in the most polluted areas. This had also been performed in the Ore Mountains, and the positive influence of fertilising and liming on forest soil and tree nutrition has been repeatedly confirmed [36,40,41,42,43]. A significant increase in pH values and exchangeable Mg content was found up to a depth of 30 cm in both the organic and mineral layers 10 years after liming; however, the increase in exchangeable Ca was significant only 2 and 5 years after liming. Despite this improvement, the low content of base cations and low base saturation of soil still prevailed in the deeper part of the soil [32].

The effects of nitrogen inputs into the forest ecosystem should also be assessed. As described in [44], the effects of nitrogen inputs into the forest range from fertilisation to acidification. These inputs influence the diversity of understorey vegetation and could negatively influence root biomass, mycorrhiza and microbial activity, and this, in turn, may negatively affect nutrient uptake, especially that of phosphorus [45,46]. Whether the ecosystem is N-limited or N-saturated depends on the load amount and duration. Our results showed a slight increase in nitrogen content in the soil and spruce needles, and the main effect was a change in the ratio between nitrogen and other important nutrients in needles, as described in Section 2.3.

## 4. Materials and Methods

### 4.1. Study Area

The Ore Mountains (Krušné hory/Erzgebirge) at the border between the Czech Republic and Saxony cover an area of about 180,000 ha. This long and rather flat-topped mountain range is formed in an NE–SW direction. It is about 130 km long and, on the Czech side, only about 6–19 km wide. The mountain plateau slopes moderately towards the NW, while the SE slopes are quite steep, dropping to the Czech brown coal basins. The degree of forestation in the Ore Mountains is about 67% [2]. Geologically, the Ore Mountains can be divided into two different parts, with the highest peak, Klínovec (1244 m a. s. l.), in the centre. The northeastern Ore Mountains are formed mainly of gneiss and granulite, with a few basalt penetrations on a limited scale (0.9%). The southwestern area is formed of phyllites and granites, which are very poor in Ca, Mg and P. The differences between the western and eastern areas of the Ore Mountains have already been described by many authors [17,47,48,49,50,51,52]. According to the taxonomic classification system of the Czech Republic [53,54], the most frequent soil types in the Ore Mountains are Podzols, represented in about 43.7% of the area, followed by Cambisols over 39.8% of the area [55]. The mean annual temperature ranges from 7.6 °C (Málkov, 370 m a. s. l.) to 2.7 °C (Klínovec, 1244 m a. s. l.), and the sum of precipitation is in the range of 514–976 mm.

The Ore Mountains, especially their northeastern parts, were exposed to a high air pollution load for a long time due to coal-burning power plants built in the brown coal basin. In the Czech Republic in 1988, annual emissions were about 2066 kt for SO_2_, 585 kt for NO_x_ and 840 kt for solid substances (dust and ash) [56]. Air pollution and deposition of acidifying elements decreased rapidly from the beginning of the 1990s after desulphurisation of power plants and new technology implementation.

### 4.2. Research Plots

The research plots were established in the Ore Mountains in young Norway spruce stands (*Picea abies* [L.] Karst) [57]. All the stands were up to 30 years old. In 1996, the monitoring system was completed, creating a transect from the southwest (plot 1 Studenec) to the northeast (plot 20 Cínovec) (Figure 16). The plots are located in the mountain ridge area at an altitude from 645 m a.s.l. to 1229 m a.s.l. The soil type is represented mainly by podzol, in three cases by stagnosoil and in one case by distric cambisol. All the plots had an area of 25 × 25 m.

The individual trees were numbered, and their defoliation and annual height increment were assessed on an annual basis [58]. 

### 4.3. Crown Condition and Growth Assessment

From 1995 onwards, defoliation of the tree crown was assessed annually at the end of the vegetation season (October–November). The defoliation of at least 30 numbered trees was evaluated within a diagonal transect in the plot, using a 5% scale, in accordance with the ICP Forests methodology [59] and modified for young Norway spruce stands [57]. These results are presented for individual plots as mean percentages of defoliation.

The height increment was measured annually in autumn in a set of 20 trees that were included in the crown defoliation assessment. The measurement was originally carried out using a Sokkia measuring pole (Sokkia, Tokyo, Japan). As the stands grew up, the method had to be changed; the Vertex hypsometer (Haglöf, Långsele, Sweden) has been used since 2006.

### 4.4. Foliage and Soil Sampling

Sampling of the needles was performed every autumn (October–November) to define their nutrient level and air pollution load, together with an assessment of the crown condition. Samples from 10 trees were collected in individual plots with 1 branch of the top part of the crown (from the 3rd to the 6th whorl) from each tree. For each plot, a pooled sample of the current-year needles and a pooled sample of one-year-old needles were created. These foliar samples were prepared in accordance with ICP Forests methodology [60]. There were 40 foliage samples analysed per year, which represents 520 samples for the entire study period.

Soil samples were taken at four-year intervals in 1995, 1999, 2003, 2007, 2011, 2015, 2019 and 2023. Individual samples of the upper organic layer and mineral soil from a depth of 0–30 cm (M03) were collected. A total of 40 soil samples were thus analysed during the individual years, and 320 during the whole study period. Sampling was carried out diagonally throughout each plot; three sub-samples were pooled prior to their analysis. The samples from the surface organic (humus) layer and mineral soil were dried at room temperature, homogenised and sieved on a 2 mm sieve prior the analysis, moreover, for analysis on CNS analyser they were ground; preparation was carried out in accordance with ICP Forests methodology [61].

### 4.5. Deposition Measurement

The deposition of elements (input with precipitation water) has been monitored in the Moldava plot in the eastern part of the Ore Mountains since 1978, when air pollution and deposition load peaked. Precipitation chemistry was measured in the forest stand (throughfall) and in the open plot (bulk deposition). Measurements started in a mature spruce stand, which was cut down due to air pollution damage in 1980. Then, a stand of mountain ash developed on this site, and in 2016, the mountain ash was underplanted with spruce. Sampling was performed twice per month until 2005, and since 2005, three times per month, according to the methodology of the ICP Forests Programme [62]. Monthly pooled samples were analysed.

### 4.6. Laboratory Analyses

Samples of the foliage, humus and mineral soil were prepared in accordance with the standard methods [60,61]. Foliage samples were decomposed in a mixture of 10 mL of concentrated nitric acid and 2 mL of hydrogen peroxide and then mineralised in a microwave oven (Speedwave 4, Berghoff). The contents of K, Ca, Mg, Al, Fe, Mn, Zn and P in needles were determined using ICP-OES. In 1995 and 1999, the nitrogen content was detected spectrometrically by Kjehldalisation, and the amounts of oxidisable carbon were determined using iodometric titration after oxidation using a chrome–sulphur mixture. Since 2003, the total C and N content has been determined using a Leco CNS elementar analyser (LECO, St. Joseph, MO, USA). The active pH(H_2_O) (in H_2_O solution) and exchangeable pH(KCl) (in 1 M KCl solution) were determined in the soil samples; the volumetric ratio of the sample in the suspension was 1:5. The pH values were measured potentiometrically using pH meter 798 MPT Titrino (Metrohm, Herisau, Switzerland) with a glass electrode. The concentration of exchangeable elements was determined in ammonium chloride extract, and the extractable content of elements was determined in Aqua regia extract using the AAS method. The available phosphorus was analysed spectrophotometrically (CFA Skalar) after being dissolved in a solution of 5 mM HCl and 2.5 mM H_2_SO_4_ in a volumetric ratio of 1.5 (sample/solution of acids). The C and N contents were determined in the same way as in needles.

Precipitation water samples were analysed as volume-weighted monthly samples, and pH values were measured potentiometrically. Conductivity was measured by conductometry, and total alkalinity was determined using titrimetric determination. Anions (SO_4_^2−^, NO_3_^−^, F^−^, Cl^−^) were measured using ion chromatography, and cations (Al, Ca, Cu, Fe, K, Mg, Mn, Na, Zn) were measured using the ICP-OES method. NH_4_^+^ and P-PO_4_^3−^ were measured using automatised spectrometry [63].

### 4.7. Data Analyses

The results for foliage and soil chemistry were compared with the classification published by [64] and with the other classification systems [65]. Prior to statistical recalculation, exploratory data analysis was carried out. A comparison was carried out with regard to the individual years during the period investigated using graphic methods (box graphs, categorised point graphs), one-factor analysis of variance and methods for comparing the independence of selected sets of data (F-test for variance, *t*-test for averages). For the correlation analysis, the Pearson R coefficient was used [66]. Nonlinear regression was used to evaluate the time trend using an element content module. All statistical data evaluations were carried out using STATISTICA software, version 13.5.0.17.

## 5. Conclusions

The Ore Mountains belong to formerly very highly polluted and damaged areas which were especially affected by sulphur and fluorine compounds. Due to high SO_2_ concentrations, which cause direct damage to foliage during severe winters, the eastern part of the mountain plateau was almost completely deforested in the 1970s and 1980s. Currently, the situation is much improved; the deposition load has decreased significantly and is comparable with other regions of the Czech Republic. The most damaged parts of the forested areas have been again afforested. Parts of the forested areas have been limed or fertilised to improve soil conditions. Although the air quality improved quickly after desulphurisation of the main pollution sources, the forest soil still remains acidified, and the base saturation of the sorption complex is still very low. The health state and nutrition level of Norway spruce stands has improved in the last three decades, and forest stands in this region seem to be stable, with defoliation lower than the national average and without serious problems with nutrition. However, the presented data identify the state of forest soils as a potentially crucial parameter for future forest development and its recovery status. In particular, phosphorus and potassium in both forest soil and foliage should remain the focus. Analysis of needles showed a slight decrease in these two nutrients and a slight increase in nitrogen concentration. As a result, the imbalance between nitrogen and phosphorus or potassium has increased.

Although the study is limited by the available long-term data series which are focused on soil quality, foliage nutrition, defoliation and height increment in Norway spruce stands, we can deduce that the lack of base cations and phosphorus may be a limiting factor for conversion of forest stands to a mixture of broadleaved and coniferous species, which are on the one hand, more stable in the changing climate and richer in biodiversity, but on the other hand, demand higher levels of soil quality. Also, the relatively high availability of nitrogen in both atmospheric deposition and forest soil, as well as the expected increase in biomass resulting from rapid forest growth, could increase the nutritional imbalances in the future. For these reasons, the monitoring of soil quality and nutrition development is recommended in the area as a base for potential planning of future liming or fertilising treatments.

## Figures and Tables

**Figure 1 plants-13-02379-f001:**
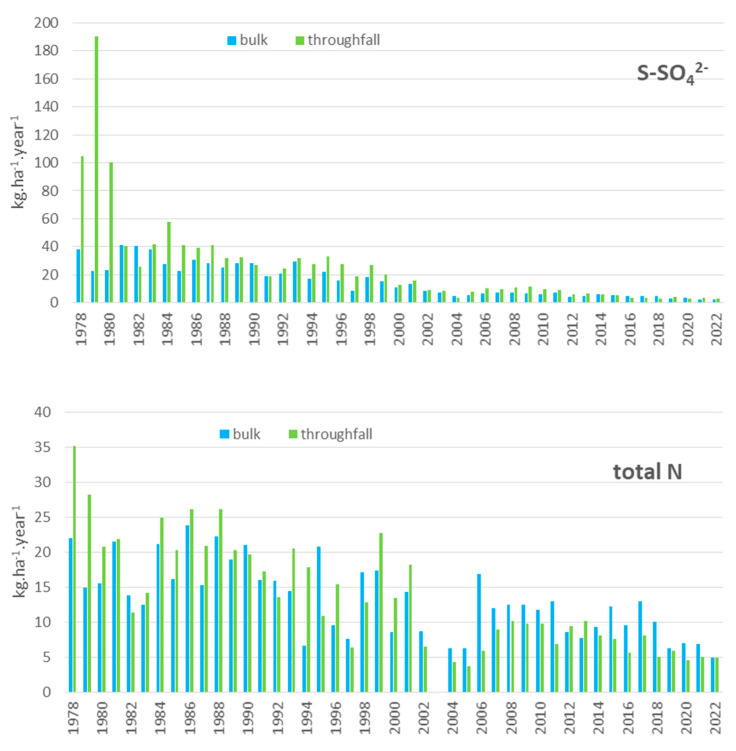
Annual deposition of sulphate (S−SO_4_^2−^) and nitrogen (sum of N−NO_3_^−^ and N−NH_4_^+^) on the Moldava plot in the Ore Mountains (kg.ha^−1^.year^−1^).

**Figure 2 plants-13-02379-f002:**
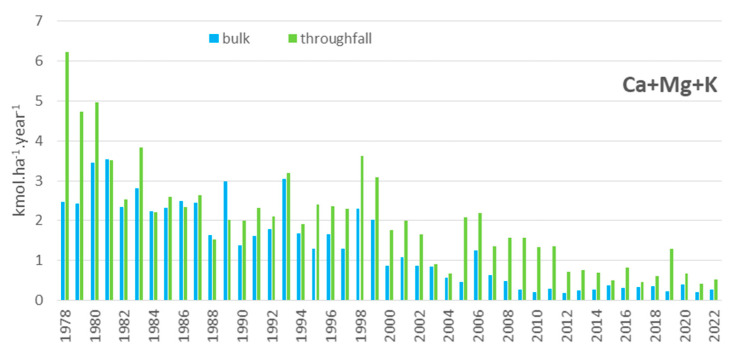
Annual deposition of base cations (Ca, Mg, K—kmol.ha^−1^.year^−1^).

**Figure 3 plants-13-02379-f003:**
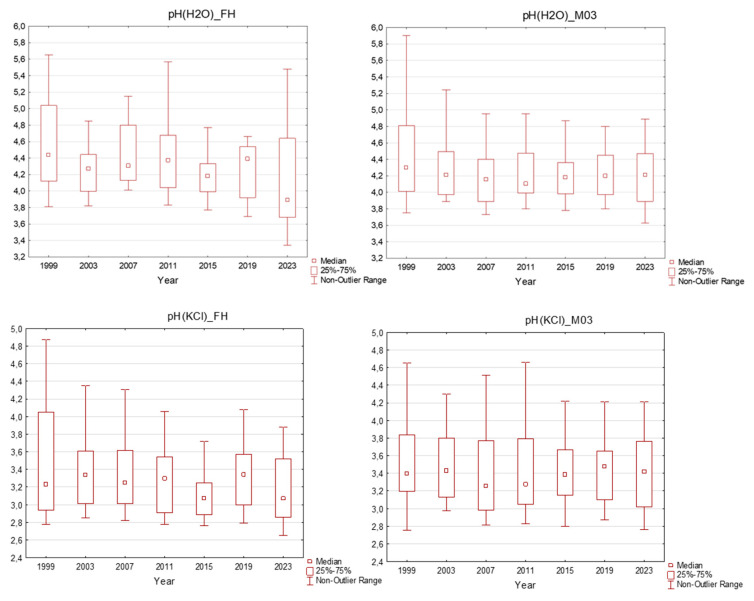
pH in the plots in the Ore Mountains. FH–organic layer; M03–mineral soil down to a 30 cm depth; middle point: median; box: 25–75%; whisker: non-outlier range.

**Figure 4 plants-13-02379-f004:**
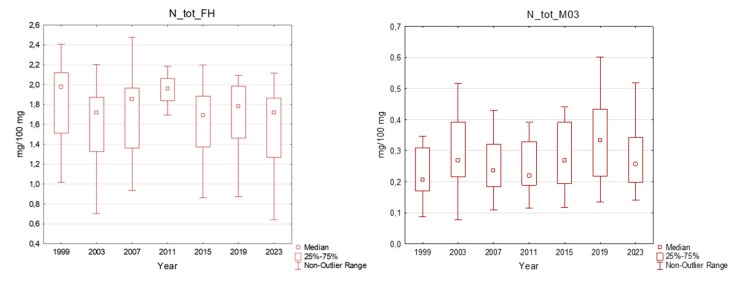
Total nitrogen content in the plots in the Ore Mountains. FH–organic layer, M03–mineral soil down to a 30 cm depth; middle point: median; box: 25–75%; whisker: non-outlier range.

**Figure 5 plants-13-02379-f005:**
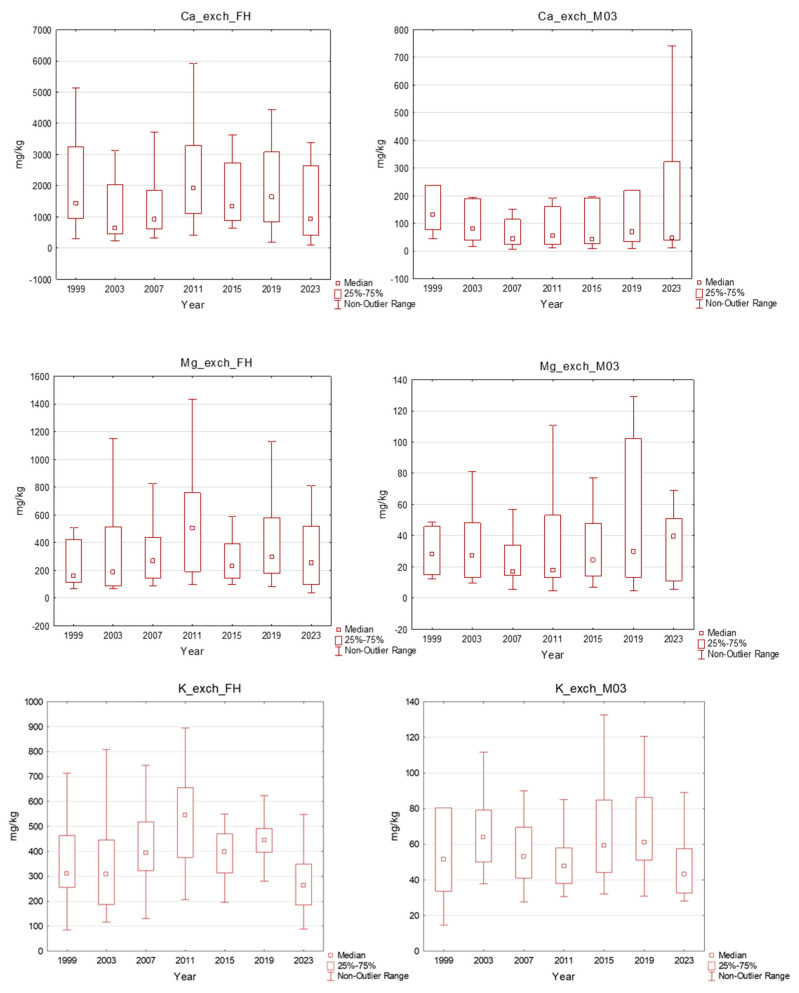
Exchangeable nutrients in the plots in the Ore Mountains. FH–organic layer, M03–mineral soil down to a 30 cm depth; middle point: median; box: 25–75%; whisker: non-outlier range.

**Figure 6 plants-13-02379-f006:**
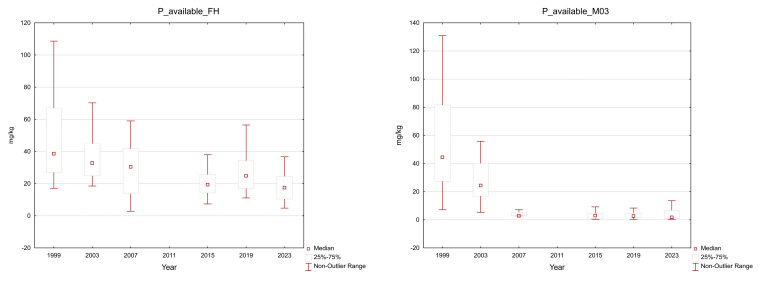
Available P in the plots in the Ore Mountains. FH–organic layer without litter, M03–mineral soil down to a 30 cm depth; middle point: median; box: 25–75%; whisker: non-outlier range.

**Figure 7 plants-13-02379-f007:**
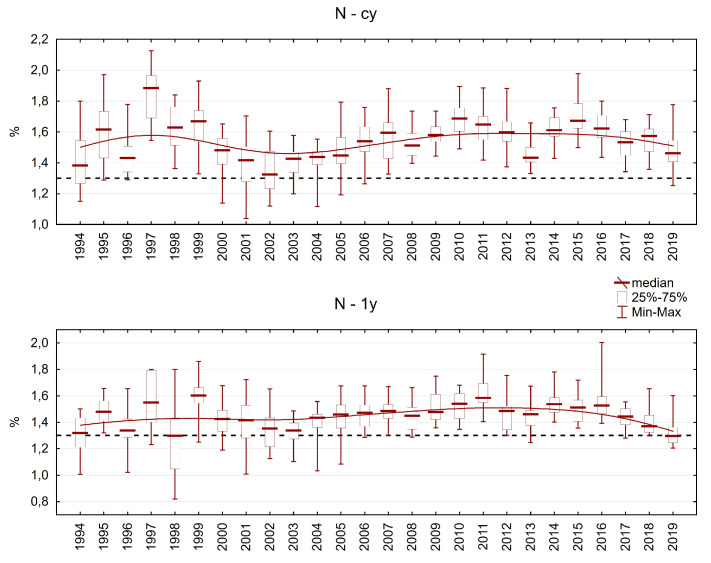
Nitrogen content in Norway spruce needles on the plots in the Ore Mountains. cy = current-year needles, 1y = one-year-old needles.

**Figure 8 plants-13-02379-f008:**
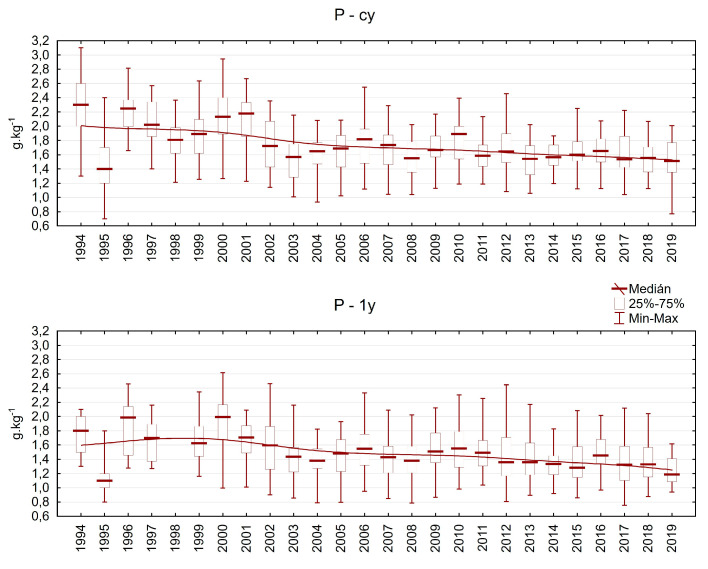
Phosphorus content in Norway spruce needles on the plots in the Ore Mountains. cy = current-year needles, 1y = one-year-old needles.

**Figure 9 plants-13-02379-f009:**
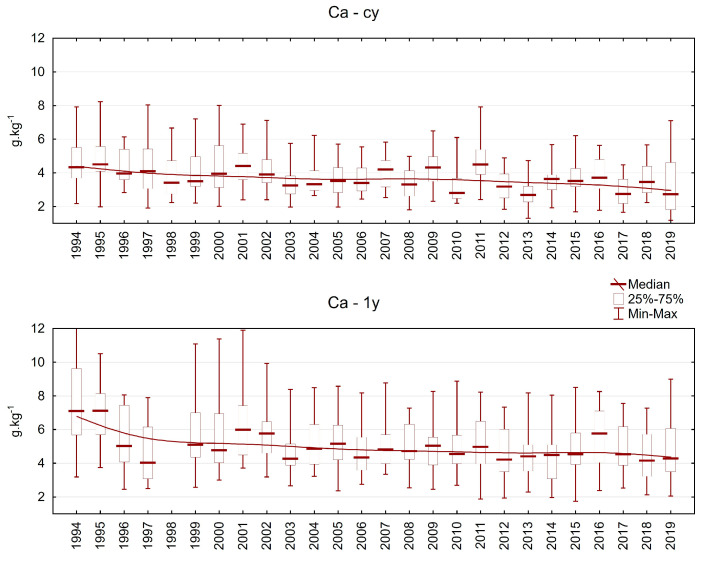
Calcium content in Norway spruce needles in the plots in the Ore Mountains. cy = current-year needles, 1y = one-year-old needles.

**Figure 10 plants-13-02379-f010:**
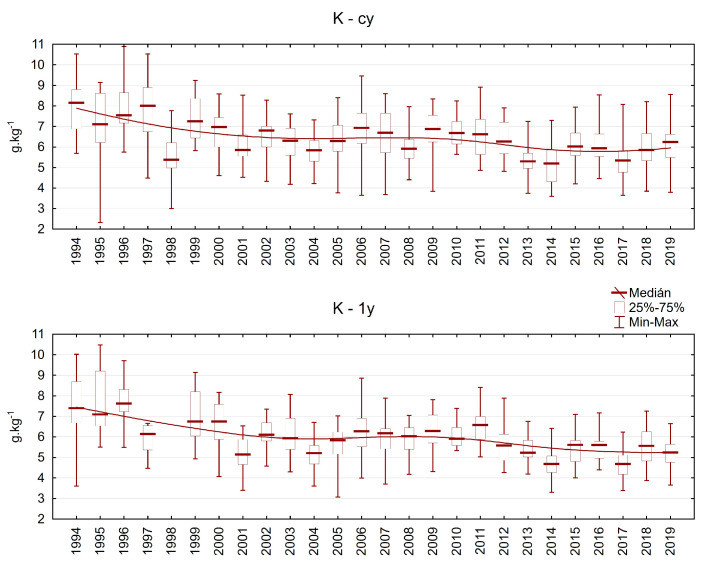
Potassium content in Norway spruce needles from the plots in the Ore Mountains. cy = current-year needles, 1y = one-year-old needles.

**Figure 11 plants-13-02379-f011:**
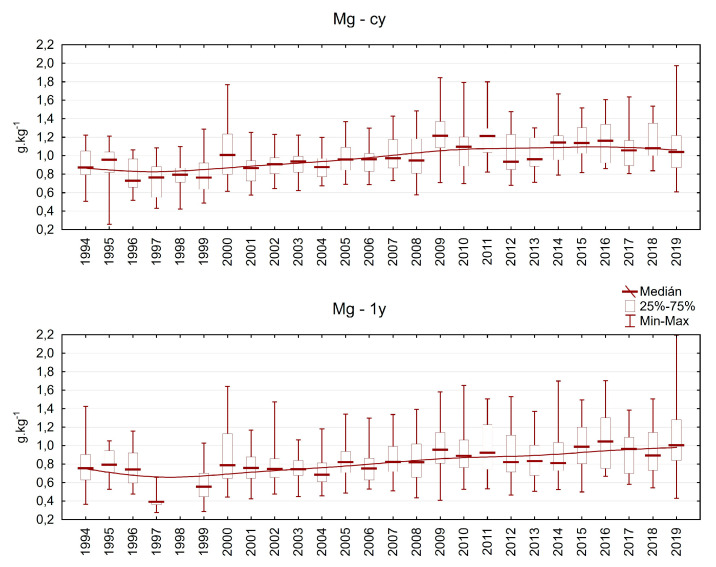
Magnesium content in Norway spruce needles from the plots in the Ore Mountains. cy = current-year needles, 1y = one-year-old needles.

**Figure 12 plants-13-02379-f012:**
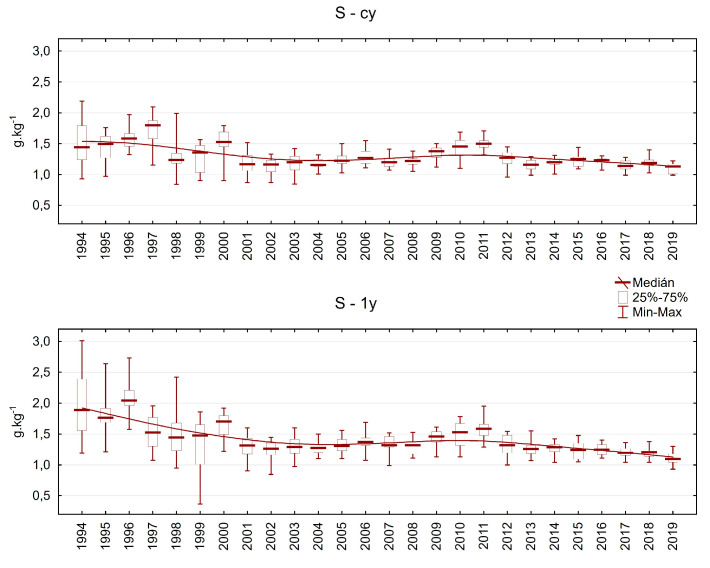
Sulphur content in Norway spruce needles from the plots in the Ore Mountains. cy = current-year needles, 1y = one-year-old needles.

**Figure 13 plants-13-02379-f013:**
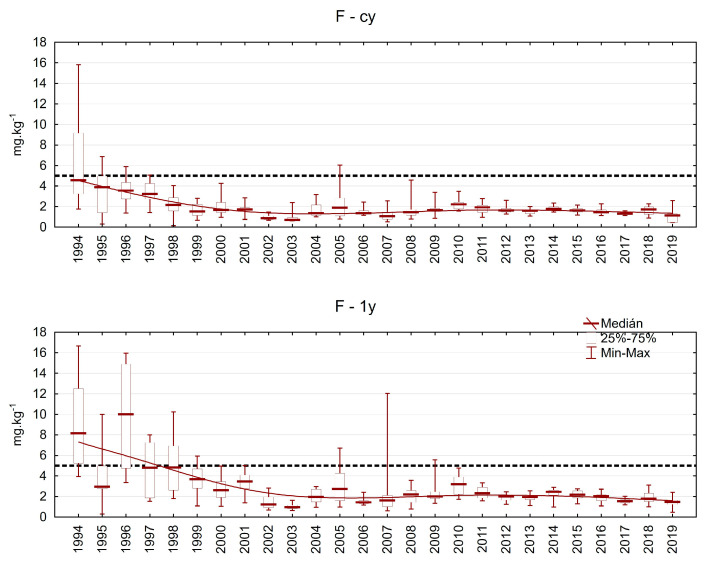
Fluorine content in Norway spruce needles from the plots in the Ore Mountains. cy = current-year needles, 1y = one-year-old needles.

**Figure 14 plants-13-02379-f014:**
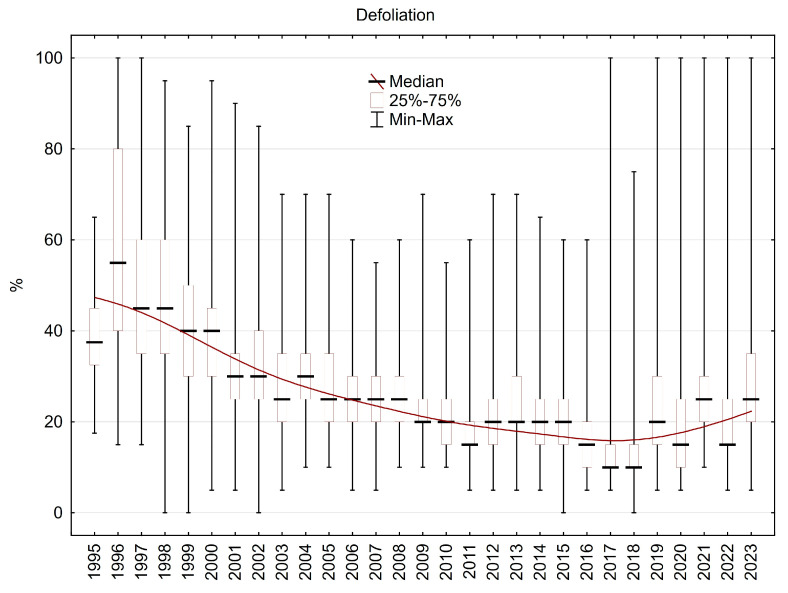
Crown defoliation of the plots in the Ore Mountains.

**Figure 15 plants-13-02379-f015:**
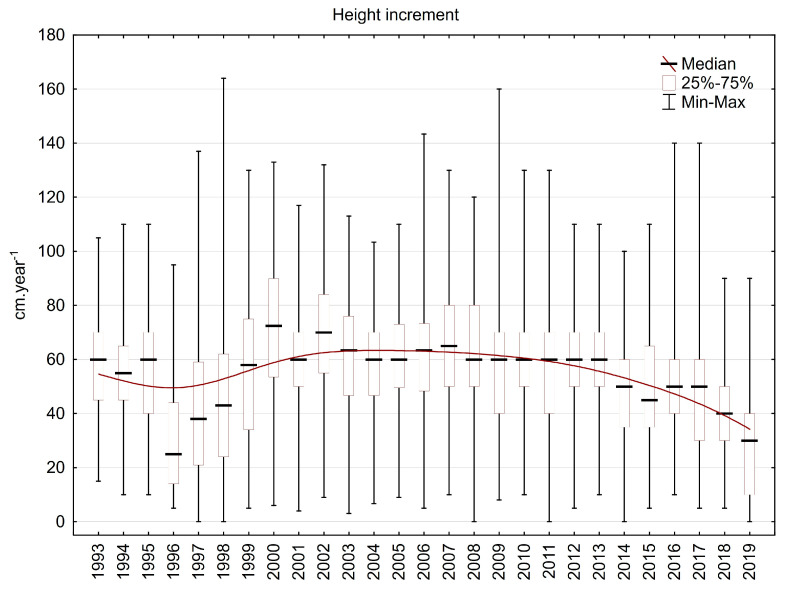
Height increment in the plots in the Ore Mountains.

**Figure 16 plants-13-02379-f016:**
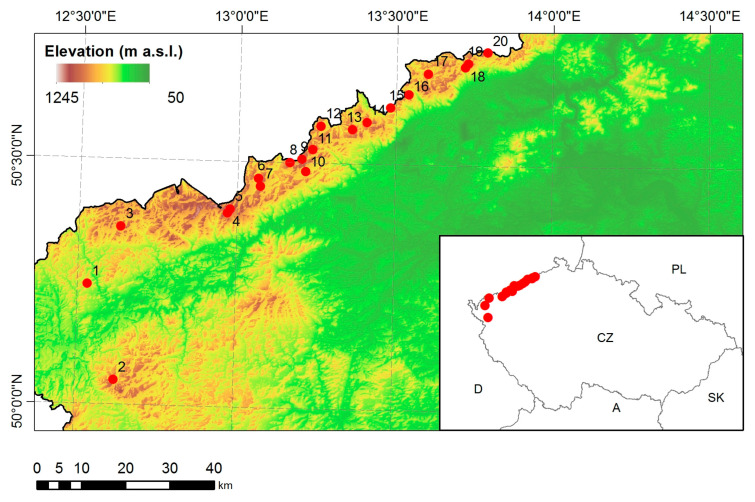
Locations of the monitored plots (Number 1–20) in the Ore Mountains.

## Data Availability

The data presented in this study are available on request from the corresponding author. The data are not publicly available due to the conditions of the Ministry—the consent of the Ministry is required for sharing or transferring data.

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
