# Peer review of "Long-Term Monitored Norway Spruce Plots in the Ore Mountains—30 Years of Changes in Forest Health, Soil Chemistry and Tree Nutrition after Air Pollution Calamity"

_plants, 2024, doi:10.3390/plants13172379_

Round 1

Reviewer 1 Report

Comments and Suggestions for Authors

Dear authors,

The manuscript ID plants-3127836, entitled "Long-term monitored Norway spruce plots in the Ore Mts.– How much time does forest regeneration take after heavy air pollution calamity? " presents a complex study with multi-year observations from 1978 to 2022.

General comments:

- chemical symbol instead of words;

- Table or figure quoted in text should be placed as close as possible (see figure 1+2);

Specific comments:

-table 1 is unnecessary, it can be as supplementary document;

- L316, Figure 1 - North position is missing;

- chapter 4.4- total number of samples is missing as well as the total quantity for soil samples; L341-342- present shortly as it is in quoted papers;

-L 355- describe the  "decomposition in microwave oven";

-L360- for pH measurement, the soil:water ratio, the type of apparatus and the manufacturer must be provided;

- L363 - the ratio sample:acids and the methodology of sample preparation for needles samples must be provided;

-L 370- typing mistake for ICP-OES;

-L 235 - Figure XX must be indicated;

-L93- references for national limits;

-L105- reference for FGMRI and it's meaning;

- page 5 half is empty; better space management by interspersing text with figures and tables to avoid empty space.

Good luck!

Author Response

General comments:

  • chemical symbol instead of words: Accepted: thank you for this comment. We changed the text – now the elements are mentioned by words when they are firstly mentioned in individual paragraphs, then the symbols are used. By more complex ions (nitrates, ammonia, sulphates) we use just symbols. If the nutrients are used in brackets (e.g. “base cations (Ca, K, Mg)” we use also symbols.
  • table or figure quoted in text should be placed as close as possible (see figure 1+2): Accepted: we try to keep this rule. In the last version of manuscript the table or figure follows its first reference on the same or next page

Specific comments:

  • table 1 is unnecessary, it can be as supplementary document: Accepted: we skip table 1 and add just general information about the soil type and altitude of individual plots.
  • L316, Figure 16 - North position is missing: Not accepted: there are meridians and parallels marked in the picture so the north position is clear.
  • chapter 4.4- total number of samples is missing as well as the total quantity for soil samples: Accepted: the yearly and total amount of samples were included in the text.
  • L341-342- present shortly as it is in quoted papers: Accepted: supplemented
  • L 355- describe the  "decomposition in microwave oven": Accepted: supplemented
  • L360- for pH measurement, the soil: water ratio, the type of apparatus and the manufacturer must be provided: Accepted: supplemented
  • L363 - the ratio sample: acids and the methodology of sample preparation for needles samples must be provided: Accepted: supplemented
  • L 370- typing mistake for ICP-OES: Accepted: corrected
  • L 235 - Figure XX must be indicated: Accepted: corrected
  • L93- references for national limits: Accepted: included
  • L105- reference for FGMRI and it's meaning; Accepted: corrected
  • page 5 half is empty; better space management by interspersing text with figures and tables to avoid empty space Accepted: the text was reorganized to fill the space efficiently.

Reviewer 2 Report

Comments and Suggestions for Authors

The study on long-term monitored Norway spruce plots in the Ore Mountains presents valuable insights into forest regeneration following severe air pollution events. However, the manuscript would benefit from a more detailed analysis of the temporal dynamics of forest recovery. Specifically, it would be helpful to include data on the specific stages of forest regeneration observed, the growth rates of both seedlings and mature trees, and any changes in biodiversity over the monitoring period. Additionally, comparisons with similar studies in other regions or with different species could provide a broader context for understanding the resilience of spruce forests to air pollution. Clarifying the impact of varying pollution levels and other environmental factors on regeneration rates would also strengthen the study’s conclusions. Kindly find some short of moderate comments to further strengthen the article.

The structure of the article does not follow the guidelines (Introduction > Material and Methods> Results > Discussion > conclusion 

Abstract needs to be written in technical ways with 1-2 lines of background information followed by objective, key findings in technical ways, with scientific recommendations 

Line no 235: Figure xx ??

Line no 276-279: need literature support to strengthen the statement (Global Ecology and Conservation, doi:10.1016/j.gecco.2020.e01302; https://doi.org/10.1080/10807039.2017.1309266)

Kindly add the future recommendation/ shortcomings / limitations of the study in the conclusion section to make the global interest to the readers  

Comments on the Quality of English Language

minor corrections only 

Author Response

  • … the manuscript would benefit from a more detailed analysis of the temporal dynamics of forest recovery. Specifically, it would be helpful to include data on the specific stages of forest regeneration observed, the growth rates of both seedlings and mature trees, and any changes in biodiversity over the monitoring period: Considered: thank you for this comment. Although it would increase the quality of manuscript, such a data or studies unfortunately are not available and we have to stick on our data sets that are presented in the paper. Considering your comment we have change a title of the manuscript to be more in line with data and results presented.
  • … , comparisons with similar studies in other regions or with different species could provide a broader context for understanding the resilience of spruce forests to air pollution: Not accepted: In discussion we compare our results with other similarly affected regions in Czechia and in Germany. All the regions strongly affected by air pollution in Europe are mountain areas with Norway spruce as the main species, so – according to our knowledge – no reliable comparison with other tree species is possible.
  • Clarifying the impact of varying pollution levels and other environmental factors on regeneration rates would also strengthen the study’s conclusions. Kindly find some short of moderate comments to further strengthen the article. Accepted: Conclusions were extended
  • The structure of the article does not follow the guidelines (Introduction > Material and Methods> Results > Discussion > conclusion: Not accepted: The order of individual chapters is according to the journal manuscript template. Papers published recently has this kind of structure with Material/Methods chapter after Results and Discussion. Please compare e.g. with: https://www.mdpi.com/2223-7747/13/15/2141 or: https://www.mdpi.com/2223-7747/13/15/2142
  • Abstract needs to be written in technical ways with 1-2 lines of background information followed by objective, key findings in technical ways, with scientific recommendations: Accepted: thank you for the comment, abstract was rewritten
  • Line no 235: Figure xx ?? Accepted: corrected
  • Line no 276-279: need literature support to strengthen the statement: Accepted: two new references were added.
  • Kindly add the future recommendation/ shortcomings / limitations of the study in the conclusion section to make the global interest to the readers Accepted: Conclusions were extended

Round 2

Reviewer 1 Report

Comments and Suggestions for Authors

The manuscript is obviously improved and if the others reviewers are agree, it can be publish in present form.

Author Response

Dear reviewer,

We would like to thank you a lot again for your revision, for all of suggestions and comments which helped to improve our text.

Best regards, authors.